# Psychometric Properties of Intercultural Competences in a Central European Context

Osama Alhendi [1], Péter Lengyel [2,*], Péter Balogh [3] and József Tóth [4]

1 Karoly Ihrig Doctoral School of Management and Business, University of Debrecen, 4032 Debrecen, Hungary; osama.alhendi@econ.unideb.hu
2 Institute of Applied Informatics and Logistics, University of Debrecen, 4032 Debrecen, Hungary
3 Institute of Statistics and Methodology, University of Debrecen, 4032 Debrecen, Hungary; balogh.peter@econ.unideb.hu
4 Institute for the Development of Enterprises, Corvinus University of Budapest, 1093 Budapest, Hungary; jozsef.toth@uni-corvinus.hu
* Correspondence: lengyel.peter@econ.unideb.hu

**Abstract:** This study aims to validate Fantini's intercultural competence scale in a sample of foreign students in a Central European context, and to figure out the pathways which are critical in improving the intercultural competence of foreign students. To achieve that, the study used confirmatory factor analysis (CFA) to construct the structural equation model (SEM). The results indicate that the scale is reliable and valid for assessing foreign students' intercultural competence. In addition, using the first-order CFA as a baseline model, the SEM indicates that each intercultural knowledge, attitude, and skills are essential in predicting the intercultural awareness of the students. On the other hand, enhanced awareness plays an important role in promoting the development of these factors. Based on that, the study provides university decision-makers with valuable information which can be helpful in formulating related policies and rules aiming to enhance the integration and intercultural contact between foreign and Hungarian students at the university environment.

**Keywords:** intercultural competence; intercultural awareness; integration; foreign students; structural equation model

## 1. Introduction

As a result of globalization and internationalization, intercultural competence becomes a very necessary requirement nowadays since it helps people to communicate with each other in effective ways, regardless of their cultural differences [1]. For example, the success of many multinational companies is attributed to the level of their employees' previous experience in the international context. Furthermore, on the level of domestic entities, addressing cultural diversity in the workplace helps to enhance efficiency. Additionally, regarding educational exchange programs, intercultural orientation is taken in consideration as a necessary factor which helps to enhance the learning of the students abroad. For instance, in certain fields such as medicine, dealing with patients with different cultural backgrounds requires sufficient intercultural knowledge which is necessary to achieve a high level of effective care. Consequently, being interculturally competent plays an important role in easing the communication across different cultures effectively [2]. In addition, various fields such as management and business [3], communication [4], linguistics [5], psychology [6], etc., take intercultural competence into account as a very important skill [7].

In this study, the paper mainly concentrates on the cultural diversity within the higher education industry. In fact, according to the authors of [8], there will be increasing demand for international higher education by students throughout the world. For example, it was predicted that the number of foreign students will increase from 1.8 million to 7.2 million during the period between 2000 and 2025. As a result, this may entail more responsibility

for preparing and helping students to perform in diverse environment effectively. This responsibility should be taken by the higher education [9]. For example, higher education usually provides students with various opportunities, such as study abroad, exchange programs, etc., which are necessary to enable them to communicate with other cultures and enhance their intercultural competence [10,11]. Furthermore, it was found out that studying abroad can help to improve different dimensions of intercultural competence such as having a good opinion of the host's culture [12], world mindedness [13], and the rise of intercultural awareness [14]. However, it is necessary to emphasize that affording study abroad programs is not enough to enhance students' intercultural competence [15]. In other words, based on the contact hypothesis, there should be a chance of interactions between students having diverse cultural backgrounds in order to minimize the level of prejudice and preconception and improve intercultural competence [16]. In addition, based on the study of [17,18], it is necessary to assess student's cognitive skills development and understand how college and the change of environment influence that. In this regard, college impact can be explained through three specific different components. The first one is student outputs which represent the development of students in terms of their knowledge, skills, attitudes, achievements, etc. The second component is student inputs such as skills, aspirations, and other possibilities for growth which students have. Hereby, inputs influence outputs and this impact can be direct or through environmental variables. The third component is college environment, related to higher education institution, which includes administrative policies and practices, facilities, as well as other characteristics which may influence the development of students. Accordingly, based on this study, intercultural competence is considered the student outputs while the University of Debrecen (UD), Hungary, is the college environment in this case.

In fact, compared to other universities, the UD has the largest international student community on the level of the country. According to Hungarian central statistical office (HCSO), the number of foreign students at the UD is about 6126, followed by the University of Pécs (4170), University of Szeged (4087), Eötvös Loránd University (3606), and Semmelweis University (3200) in the academic year 2019/2020. In addition, as shown in Figure 1, the number of foreign students at the UD has gradually increased during the period between 2014 and 2019 [19].

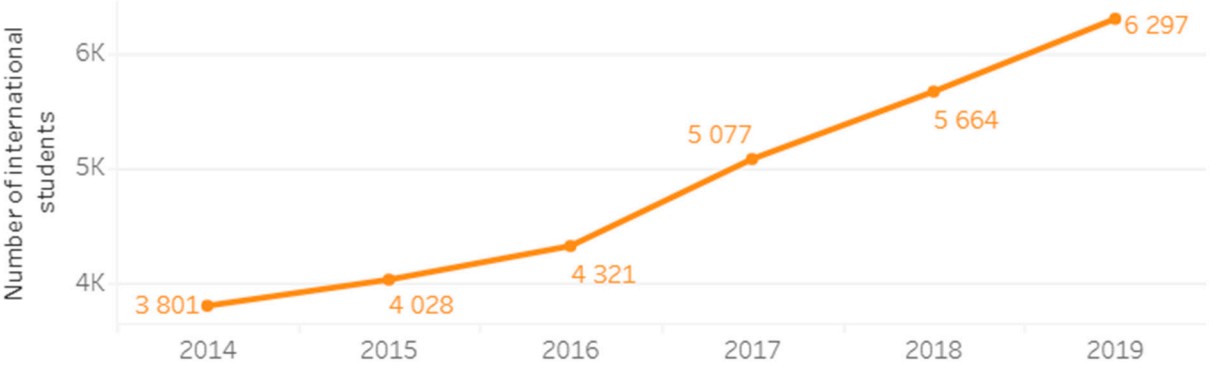

**Figure 1.** The number of international students at the University of Debrecen, 2014–2019.

Moreover, this rise can be interpreted through the following facts. Firstly, since 2000, Hungarian universities have experienced many structural as well as strategic changes. These changes have contributed to the process of transformation of many important aspects such as the governance, the scope, and the structure of the international and internal mission and activities of the higher education institutions [20]. Secondly, in 2013, the Hungarian government launched a scholarship program called Stipendium Hungaricum. The main aim of this program is to internationalize Hungarian higher education and enhance its development [21]. Hereby, besides Erasmus+ and Campus Mundi programs,

Stipendium Hungaricum results in a high level of cultural diversity on the level of the nation. As a result, the previously mentioned universities, which have a large quota of foreign students, experience high levels of diversity too.

In this paper, the study focuses on the cultural diversity at the UD. In fact, the UD is considered one of the biggest education centers in the level of the country [22]. In addition, it has the largest quota of foreign students. Hereby, there is a high level of cultural diversity. For example, Figure 2 shows the nationality as well as the number of foreign students, by nation, who study at the UD. Based on the figure, there is a wide range of nationalities, cultures as well as languages in the university environment. According to [23], there were about 3077 Asians, 1519 Europeans, 1279 Africans, 246 students from Latin as well as North American continents, and 5 from Oceania.

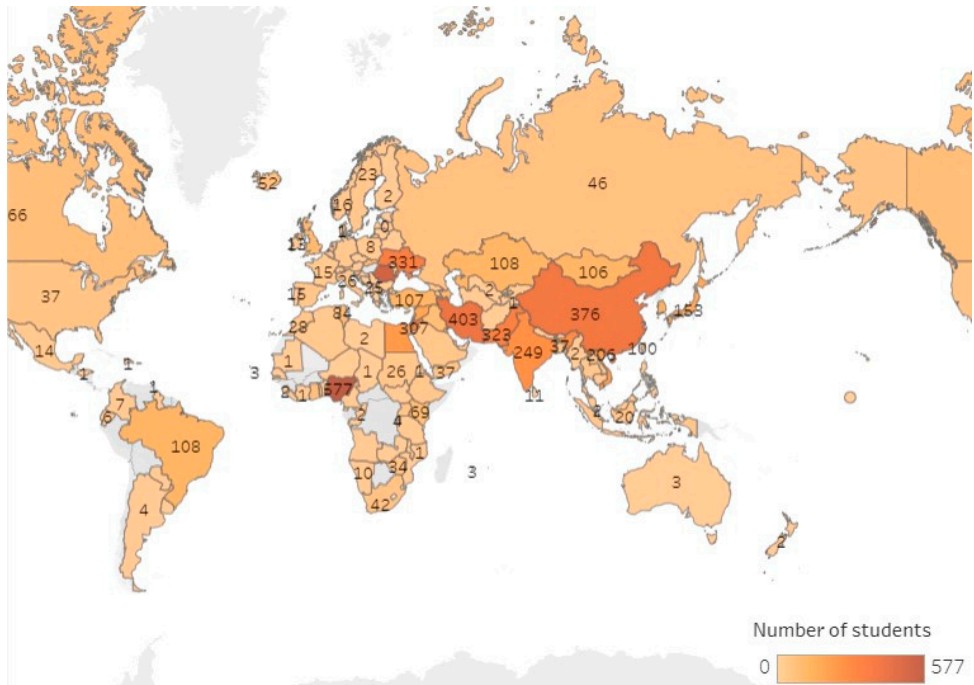

**Figure 2.** The number of international students by country—University of Debrecen, 2019.

Consequently, as a response to these changes, the university's student services should be re-evaluated and improved in terms of their suitability and appropriateness. This procedure could enable the university to be more responsive to the requirements of domestic as well as foreign students. Furthermore, the authors of [24] argue that the academic progress of students is highly affected by their engagement and integration. Therefore, the university should take this issue into account.

In reality, most of the literature focuses on the obstacles and challenges which foreign students experience in the Hungarian higher education system [25,26]. However, there is not enough literature focusing on the foreign student's integration into the new college environment [27]. In addition, it was found that foreign students have a higher willingness and readiness to communicate with Hungarian students. On the other hand, Hungarian students are hesitant to contact foreign students [28]. Both Hungarian and foreign students attribute the problem of lack of communication to the language barrier [27]. Hereby, Hungarian universities should adopt the integrative approach in order to enhance their student's intercultural competence and integration.

Based on the previous facts, this study aims to investigate foreign students' intercultural competence at the UD. In order to achieve that, the study uses Fantini's intercultural competence model [29]. This model is suitable for this study because it concentrates on foreign students experience abroad and helps to assess students' intercultural competence

in terms of four related dimensions (i.e., intercultural knowledge, attitude, skills, and aware­ness). Furthermore, the study contributes to the literature on foreign students' experience in Hungary, especially at the UD, in different ways. Firstly, the literature on foreign students' intercultural competence levels at the UD is insufficient. Secondly, based on the results of this study, intercultural knowledge, attitude, skills, and awareness play an important role in enhancing foreign students' intercultural competence at the university environment. Thirdly, the study provides important feedback from respondents. This feedback can be considered valuable information by which university's decision makers can formulate the effective policies and procedures in order to enhance the level of integration between Hungarian and foreign students in the university environment. Furthermore, this can also raise the level of intercultural competence of the two groups.

Based on the literature, it was found that numerous adult students lack the neces­sary intercultural skills [30,31]. Moreover, the lack of intercultural competence results in several problems such as prejudice, discrimination, and unfriendly speech. This could happen because of misunderstandings and miscommunication between people of diverse cultures [32,33]. Hereby, it is necessary to investigate the means and activities which help to enhance intercultural competence among students. To achieve that, it is necessary to first provide a detailed clarification of the theoretical framework of intercultural competence.

At the beginning, it is noteworthy to shed the light on the difference between inter­cultural competence and intercultural communicative competence (ICC). The first term is defined as "the ability of individuals to interact in their own language with the people from another country and culture". On the other hand, ICC includes language teaching as well as "the ability to interact with people from another country and culture in a foreign language". Furthermore, people with developed ICC are efficient in communicating with others from different cultures [34,35]. In addition, the development of ICC can be enhanced through intercultural communication. For example, it was argued that direct communi­cation with native people helps to increase the intercultural skills and attitudes of foreign students [36,37]. Additionally, based on the experience of Chinese exchange students in United Kingdom, it was found out that exchange programs raise students' intercultural sensitivity [38].

In fact, after five decades, reaching an agreed definition of intercultural competence is still not achieved [39]. Based on the literature, the majority of ICC models have tried to include specific individual traits, in which individuals' skills and attitudes can be attributed to the measurement of successful intercultural behavior. This may include intercultural adaptation, appropriateness, and effectiveness of the communication [1]. Furthermore, it is possible to argue that differences could exist among the theoretical frameworks of intercul­tural competence. However, there are correspondences too [40]. For instance, ref. [41] has determined 264 items of intercultural competence models and theories, in which there are 127 behavioral and skill factors, 77 attitudinal dimensions, and 64 cognitive traits [42]. In other words, whereas some authors, such as the authors [43,44], adopted a cross-cultural attitude approach, in which ICC is defined as the ability of individuals to show a posi­tive attitude toward other cultures, others, such as the authors of [45,46], employed the behavioral skills approach. These skills include displays of respect, interaction posture, orientation to knowledge, empathy, self-oriented role behavior, interaction management, and tolerance for ambiguity.

On the other hand, there are many models which are specifically used to estimate the intercultural competence of students. The most appropriate models which can be used in the higher education industry are the Intercultural Development Inventory (IDI), the Cross-Cultural Adaptability Inventory (CCAI), Cross-Cultural World-Mindedness Scale (CCWMS), Intercultural Sensitivity Inventory (ISI), and the Assessment of Intercultural Competence (AIC) [47].

Regarding IDI, it was created by the authors of [48]. It can help to explain and understand the experience of people toward other cultures. Additionally, this model was developed based on the Developmental Model of Intercultural Sensitivity (DMIS)

which is proposed by the authors of [49]. The DMIS includes six stages, in which the individuals develop from the stage of ethnocentrism (denial, defense, and minimization) to ethnorelativism (acceptance, adaptation, and integration). Both IDI and DMIS can be used in the study abroad programs, related workshops, and curriculum design [50].

Regarding the Cross-Cultural Adaptability Inventory (CCAI), it is developed by the authors of [51]. It aims to evaluate the effectiveness of people in cross-cultural situations. It includes specific factors such as emotional resilience, flexibility and openness, perceptual acuity, and personal autonomy. It is useful in assessing intercultural training programs for certain groups of people such as graduate students [52], business and medical specialists [53], and people who learn languages [54].

With respect to the Cross-Cultural World-Mindedness Scale (CCWMS), it is a survey including 26 items, which is created by the authors of [55] and based on the previous study by the authors of [56,57]. It aims to assess people's attitudes toward race, religion, global education, etc. In addition, the concept of world-mindedness indicates an individual's "positive attitudes toward issues such as immigration, world government, and world economic justice" [58].

As for the Intercultural Sensitivity Inventory (ISI), according to the authors of [59], it helps to evaluate the ability of individuals to appropriately adjust their behavior while he/she communicates with other cultures. Furthermore, the construct of this instrument includes individualism, collectivism, flexibility, and open-mindedness.

The following paragraphs provide a detailed explanation of the fifth instrument since the study considers it as an appropriate tool, compared to the above-mentioned models, for investigating the intercultural competence level of foreign students at the UD.

The Assessment of Intercultural Competence (AIC), made by the Federation of the Experiment in International Living (FEIL), is an endeavor to evaluate the intercultural results of its projects. Hereby, FEIL specialists define intercultural competence as "a complex of abilities needed to perform effectively and appropriately when interacting with others who are linguistically and culturally different from oneself" [29].

According to Fantini, intercultural competence consists of four interrelated dimensions (intercultural knowledge, intercultural attitude, intercultural skill, and intercultural awareness). These dimensions are well known as the KASA acronym (see Figure 3). Regarding knowledge and skills, it is possible to address them in usual educational settings. Additionally, it is easy to assess them since they are measurable (by grades or numbers). As for attitude and awareness, people having intercultural experience believe that these two dimensions are crucial and critical for the intercultural success [60].

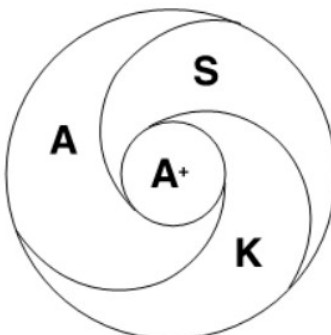

**Figure 3.** A+ASK quartet.

Based on the KASA quartet, intercultural knowledge indicates how much the individuals (students) are familiar with their own culture as well as other cultures that they communicate with. Furthermore, Fantini includes the ability of students to collect the necessary information as a necessary factor by which communication can be more successful [29]. Secondly, intercultural attitude refers to a student's attitude toward other cultures. This may include openness and showing respect, interest, and curiosity about other cul-

tures [61]. Additionally, risk orientation, as well as empathy, are other forms of intercultural attitude [62]. The next dimension is intercultural skill. This dimension includes various skills such as the student's ability to listen, analyze, and compare his/her own culture with other cultures [61]. Other skills could also include the ability to learn language of other cultures, distinguish and compare different languages, solve the problems, and collect information [62]. The fourth dimension is intercultural awareness. Based on the study in [34], intercultural awareness indicates the ability of individuals (students) to understand and notice the similarities and differences between their own cultures and other cultures. Furthermore, as observed in Figure 3, awareness is placed in the center as an important dimension. In other words, awareness can be strengthened through the development and improvement of knowledge, attitude, and skills. Additionally, at the same time, enhanced awareness helps to further promote their development [27,63]. According to the work of [64], awareness is raised through introspection and reflection which are integral part of intercultural experience. Based on that, Fantini defines awareness as "in and of the self and it is always about the self in relation to someone or something else". Furthermore, awareness is one of the most cited dimensions in the literature.

Besides these four dimensions, Fantini considers the proficiency of the host's language as a necessary component of ICC. Learning a second language helps to enhance intercultural communication. On the contrary, individuals who speak their own languages cannot access essential aspects of the host's world [64].

The reasons behind choosing Fantini's model as a part of this study can be justified by the following. Firstly, compared to other previously mentioned models, Fantini has assured that each dimension consists of many several items which are critical for the success of intercultural communication. Secondly, the dimensions are interrelated and linked, and this results in the creation of complex competence. Thirdly, Fantini's model has been used in different studies focusing on student mobility as well as skilled immigrants. Hereby, this provides evidence of the model's operationalization in this field [5]. Fourthly, unlike other models, Fantini has included awareness as a central component of the KASA acronym of intercultural competence. For example, to some extent, Byram's model [5] consists of similar dimensions (knowledge, skills, and attitudes) as well as critical cultural awareness. However, whereas Byram work targets schoolchildren, Fantini's model focuses on young adult individuals in the host country. As for the work of Ang and Van Dyne's [65], their model includes knowledge, motivation, behavior, and strategy. Therefore, awareness is not considered a distinguished component since motivation, here, indicates the willingness and interest of individuals to interact with and experience other cultures. Similarly, Ref. [66] does not regard awareness as separate dimension. Instead, awareness is treated as a part of knowledge and comprehension component. The rest of dimensions are attitudes, skills, desired internal outcomes, and desired external outcomes.

By looking at the literature on Hungarian higher education, there are not enough studies that concentrate on the validity and reliability of Fantini's intercultural competence scale for foreign students at the UD. Based on that, the study takes the advantage to investigate that by addressing the following questions:

1.  Does awareness of foreign students play a critical role in the achievement of successful intercultural interaction?
2.  Is the language required in the intercultural communication between foreign and domestic students in the university environment?
3.  What are the policies and procedures the UD should adopt in order to enhance the intercultural competence level of its students?

This study contributes to the existing literature on intercultural competence as follows. Firstly, by looking at the Hungarian literature, this study is considered one of the few research projects that use the intercultural competence theory to study the function of intercultural competence and common language within Hungarian higher education. Secondly, the model of Fantini and his theory was successfully validated and confirmed in a new environment (Central Europe). Thirdly, the study provides several policies that

are necessary to address the obstacles which hinder the internationalization process of Hungarian universities. Fourthly, it emphasizes the importance of the integrative approach, which is missing in most Hungarian universities at the current time. Therefore, the culture of Hungarian institutions should be changed to enhance the intercultural competence of international and Hungarian students as well.

## 2. Theoretical Framework

By looking at the literature, it is worth mentioning that it is not easy to identify a certain theory that can be regarded as a leading one in the field of intercultural competence. However, there are a lot of different theories in the literature that could help explain intercultural competence very well. For example, based on the scientific work of Berger and Calabrese [67], Gudykunst [68] was able to develop the theory of Anxiety Uncertainty Management in which he argues that intercultural communication competence can be established by two important factors: effective anxiety management and a person's ability to be mindful. On the other hand, the theory of Ting-Toomey [69], Face Negotiation Theory, claims that people strive to preserve a positive social self-image. Therefore, based on this theory, competence is a part of the concept of face, which can be attained through the combination of an individual's knowledge, mindfulness, and skills while communicating with others. Furthermore, there is another set of theories which could explain intercultural competence from different points of view (systems perspective). For instance, to define the ICC, Spitzberg [70] has developed and determined three different levels. The first level is called the individual system. At this level, competent communication can be eased and facilitated through specific characteristics that individuals may have. The second one is called the episodic system. According to this level, in a certain episode of communication, there are several features an actor should have to influence the co-actor's impressions. The third level is the relational system, in which there are specific components or variables that supports individuals' competence across all their relationships rather than simply one episode of interaction. On the other hand, Cultural Adaptation Theory, which was developed by Kim [71], argues that the purpose of each person, as an open system, is to adapt to one's surroundings. In addition, according to this theory, ICC consists of three different dimensions: cognitive, affective, and behavioral attributes. As for the cognitive aspect, Chen [72] defined it as intercultural awareness. It includes the ability of the person to understand and explain verbal as well as non-verbal messages. In order to achieve that, individuals are required to have the necessary knowledge about host culture's values, norms, beliefs, etc. On the other hand, the affective dimension is defined as the ability of people to empathize the emotional experiences of the members of the host culture and accept the cultural differences. Finally, the behavioral component is concerned with one's capacity to operate in the host community [73]. Based on the previously mentioned theories, it is possible to define intercultural competence as a combination of capabilities to engage in different cultures and situations effectively and appropriately [42,60,74,75].

This study mainly concentrates on the intercultural competence of foreign students in the higher education industry. Based on the literature, there are many different theories that address this issue (as observed in Table 1). These theories were previously defined and explained at the end of Section 1.

**Table 1.** ICC models used in the higher education industry.

| ICC Models | Dimensions | Authors |
|---|---|---|
| Intercultural Development Inventory (IDI) | Denial/Defense, Reversal, Minimization, Acceptance/Adaptation, and Encapsulated Marginality. | Hammer et al. [48] |
| Cross-Cultural Adaptability Inventory (CCAI) | Emotional Resilience, Flexibility and Openness, Perceptual Acuity, and Personal Autonomy. | Kelley and Meyers [51] |
| Cross-Cultural World-Mindedness Scale (CCWMS) | It aims to assess people's attitudes toward race, religion, world government, war, patriotism, and global education. | Der-Karabetian [55] |
| Intercultural Sensitivity Inventory (ISI) | Individualism, collectivism, flexibility, and open-mindedness. | Bhawuk and Brislin [59] |
| The Assessment of Intercultural Competence (AIC) | Knowledge, Attitude, Skills, and Awareness (KASA). | Fantini [29] |

Source: [47].

In this study, the intercultural competence of international students is investigated by the AIC tool, which was developed by Fantini. According to Fantini [29], intercultural competence is "a complex of abilities needed to perform effectively and appropriately when interacting with others who are linguistically and culturally different from one's self". As previously mentioned, Fantini argues that ICC can be identified through intercultural awareness, attitude, skills, and knowledge. As for an individual's intercultural skills and knowledge, it is possible to estimate them. On the other hand, having a positive intercultural attitude and awareness is helpful and necessary to achieve intercultural effectiveness.

This model is relevant to this study due to the following justifications. For instance, Fantini's model and its dimensions have all components which are necessary for intercultural effectiveness. Additionally, intercultural knowledge, skills, attitude, and awareness are interconnected and help to form complex competence. Furthermore, according to the literature, the model of Fantini was used by many previous studies addressing topics related to student mobility and immigration. Based on that, this model could provide useful information in this area [76]. Consequently, using the theory of Fantini, in which intercultural skills, attitude, and knowledge have positive effect on intercultural awareness and vice versa, it is possible to investigate these interrelationships and determine which dimension requires careful attention compared to the others. For instance, having sufficient knowledge about other people's cultures enhances an individual's intercultural awareness [77]. Furthermore, the intercultural awareness of individuals can be boosted by the extent to which a person has a positive attitude toward different cultures [78]. In addition, there is a positive relationship between individuals' intercultural skills and awareness [79]. Finally, according to Byram [5], intercultural awareness represents a person's capacity to have knowledge about and become more tolerant toward different cultures.

Based on the Hungarian literature, it was found that there is a lack of communication between Hungarian and foreign students. This lack of interaction was justified due to the introverted personality of Hungarian students as well as linguistic anxiety which makes them hesitant to socialize with other cultures [27,80,81]. In addition, based on the study of Lannert and Derény [82], the internationalization of Hungarian higher education is mainly influenced by the internal culture of the institution as well as the misconception of leadership. Therefore, these current problems require careful attention from Hungarian universities. In other words, the university environment (e.g., administration policies) should adopt new approaches to enhance student output (e.g., intercultural competence and other skills) [17,18]. In return, these issues could negatively affect the intercultural

experience of foreign students during their stay in the host culture. As a result, this study tries to investigate and test foreign students' intercultural competence (student output) and their attitude toward these issues. Furthermore, since intercultural competence is a necessary requirement in the labor market, this study aims to use AIC tool in order to provides us with valuable information on foreign students' intercultural competence, in terms of their intercultural awareness, skills, knowledge, and attitude. In addition, the tool provides space for foreign students to highlight their attitude toward the host culture and illustrate their feedback and needs which are necessary for policy development.

## 3. Materials and Methods

### 3.1. Participants

The target population was the foreign students at the UD. The participants of this study were randomly chosen. To ensure representatives, the sample size was 384 foreign students. Among the respondents, there were 221 (57.4%) males and 163 (42.3%) females. Regarding foreign students' continents, the sample size consists of 213 (55.3%) Asian students, 60 (15.6%) Europeans, 79 (20.5%) Africans, and 32 (8.3%) students from Latin and North America continents.

As for the time spent abroad, it ranges between more than two years (167 foreign students (43.3%)), more than one year (102 (26.5%)), and one year or less (114 (29.6%)).

### 3.2. Instruments

The survey includes three sections. The first one includes the sociodemographic items such as nationality, native language as well as the ability to speak other languages, gender, educational level, and time spent in Hungary. Besides that, it was necessary to identify the linguistic challenges which foreign students could face during their stay in Hungary. Hereby, specific item statements (e.g., did you go on to study/learn Hungarian language? Did you face linguistic difficulties to communicate with Hungarians?) were added to the survey.

In the second section, the intercultural competency of foreign students was estimated using the Assessment of Intercultural Competence (AIC) which was developed by Fantini [29,83]. The scale consists of 53 items with a 6-point Likert-type scale (ranging from 0 = Not at all to 5 = Extremely high). Based on Fantini model, KAS+A acronym includes the following subscales: Intercultural Knowledge (items from 1 to 11), Intercultural Attitude (items from 12 to 24), Intercultural Skills (items from 25 to 35), and Intercultural Awareness (items from 36 to 53).

In the third section, the last item was an open-ended question, which has provided foreign students with possibility to share their thoughts and ideas regarding their intercultural experience at the UD in Hungary.

### 3.3. Procedures

Due to the COVID-19 pandemic and the announced lockdown on the 14th of November 2020, the Hungarian education system was switched to online teaching. Hereby, it was not possible to directly circulate the survey among foreign students. However, it was feasible to create an online survey and share its link through UD's online platforms, teacher's help, direct messages, as well as related social media groups. Survey circulation started from December 2020 until the middle of April 2021.

### 3.4. Data Analysis

The analysis of the data was conducted using SPSS 24. On the other hand, SPSS Amos 24 was used to carry out confirmatory factor analysis (CFA) and structural equation modeling (SEM).

Exploratory factor analysis (EFA) was first used to define the nature of the relationship between factors. In the beginning, data suitability for factor analysis was checked through Kaiser–Meyer–Olkin (KMO) (cut-off value of 0.60) and Bartlett's Test of Sphericity (where

$p < 0.05$) [84]. To identify which rotation method should be used, principal component analysis (PCA) with oblique rotation was conducted. Accordingly, using the component correlation matrix, it is possible to use the oblique rotation method when the values are higher than $\pm0.32$ [85].

As for confirmatory factor analysis (CFA), using maximum likelihood (ML), the model fit was checked through the goodness of fit indices and their cut-off values as follows, root mean square error of approximation (RMSEA) < 0.06, comparative fit index (CFI) > 0.95, standardized root mean squared residual (SRMR) $\leq$ 0.08, and minimum discrepancy function by degrees of freedom divided (CMIN/DF) should be between 1 and 3 [86]. Based on CFA results as well as validity and reliability, it is possible to conduct structural equation modeling for Fantini's KASA variables since awareness can be enhanced through the development of knowledge, skills, and attitude.

Furthermore, intercultural competence is considered a general factor in this study. Therefore, it is possible to introduce second-order CFA since the second-order factor (intercultural competence) can be specified to account for the covariances among the three first-order factors (knowledge, attitude, skills) [87].

In relation to the reliability of the factors, it was estimated using Cronbach's alpha, in which $\alpha$ of 0.7 is acceptable [88] and indicates that items are adequate in assessing the constructs of intercultural competence. Furthermore, convergent validity [89], indicated by composite reliability (CR) and average variance extracted (AVE), as well as discriminant validity [90], are estimated by comparing AVE with maximum shared squared variance (MSV).

## 4. Results

In line with the methodology, the results are accordingly organized as follows.

### 4.1. Descriptive Statistics

Regarding the first section, sociodemographic items, the concentration was on the language proficiency of foreign students. Based on the answers to the question, e.g., "did you go on to study/learn Hungarian language?" about 50.90% of the respondents confirmed by answering yes. Furthermore, about 87.24% of the respondents confirmed that they have faced linguistic difficulties to communicate with Hungarians.

### 4.2. Exploratory Factor Analysis (EFA)

Before conducting factor analysis, it is important to check the suitability of the data for factor analysis, using Kaiser–Meyer–Olkin (KMO) and Bartlett's tests. Hereby, according to the results, KMO = 0.968 (higher than 0.60). Additionally, the $p$-value of Bartlett's tests is 0.000 ($\chi^2$ = 19082.359). Therefore, the data are suitable for the factor analysis.

As for rotation type, based on the component correlation matrix, the values were higher than $\pm0.32$. As a result, it is recommended to use the oblique rotation method. In this study, EFA was conducted using maximum likelihood extraction with direct oblimin rotation. Furthermore, four factors were identified with an eigenvalue higher than 1.00. For instance, the first factor's eigenvalue is 25.899 and it explains about 48.866% of the variance. The second factor's eigenvalue equals 4.295 which explains 8.103% of the variance. As for the third and fourth factors, their eigenvalues are 2.634 and 1.939 respectively (and they explain about 4.969% and 3.659%) and together explain a cumulative variance of 65.598%. In relation to the pattern matrix, using extraction and rotation with an absolute value of 0.5, it was recommended to remove attitude item No. 5 and skills item Nos. 1, 2, and 3. Therefore, items T1 to T4 and T6 to T13 load on a single factor (attitude which is indicated by T). Items S4 to S11 load on a single factor (skills, indicated by S). Items K1 to K11 load on a single factor called knowledge in this case (indicated by K). Finally, items A1 to A18 load on a single factor called awareness (indicated by A).

As for internal consistency, Cronbach's alpha of attitude factor equals 0.961 (which is higher than the cut-off value of 0.80 according to [91]). Regarding skills, knowledge,

and awareness, Cronbach's alpha of each factor equals 0.938, 0.934, and 0.968, respectively. Besides that, K6 has the highest mean (3.15625), as the most important item compared to the rest, in the knowledge scale. Regarding attitude, skills, and awareness scales, T7, S1, and A1 (3.46875, 3.61197, and 3.60677 respectively) have the highest means in comparison with other items.

*4.3. Confirmatory Factor Analysis (CFA)*

4.3.1. First-Order CFA

In this stage, it is possible to improve the model using modification indices. Based on the theory, nine modifications were performed. As a result, the model's goodness of fit was improved (as observed in Table 2). Furthermore, the first-order CFA is shown in Figure 3.

**Table 2.** Goodness of fit indices—first-order CFA.

| Fit indices | Value | Interpretation |
|---|---|---|
| Chi-square/*p*-value | 2567.394/*p* = 0.000 | |
| DF | 1112.000 | |
| CMIN/DF | 2.309 | Excellent |
| RMSEA | 0.058 | Excellent |
| CFI | 0.916 | Acceptable |
| SRMR | 0.042 | Excellent |

Note: as previously mentioned, the cut-off values for goodness of fit are the following. RMSEA < 0.06, CFI > 0.95, CMIN/DF, between 1 and 3, and SRMR ≤ 0.08 [86].

Therefore, Table 1 indicates that the first-order CFA has satisfactory goodness of fit indices. Additionally, Figure 4 shows standard regression values for each item. Consequently, the results of CFA confirm the fitness of the factors. Hereby, the data of this study fit a hypothesized measurement model.

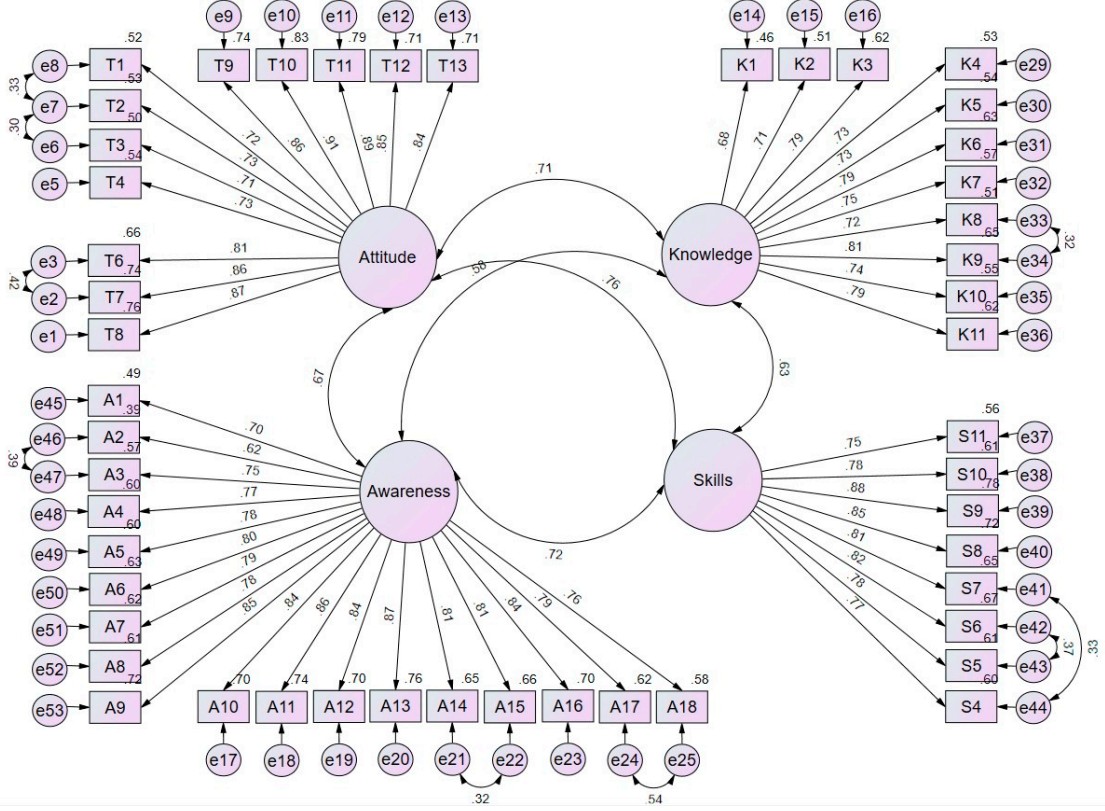

**Figure 4.** First-order CFA (*n* = 384) with standardized estimates.

In addition, as observed in Table 3, convergent and discriminant validity were checked. As for AVE, all the values for the four dimensions are higher than 0.5. Therefore, the model has good convergent validity, which means that every item really converges to the assessment of its related construct. Furthermore, all values of CR are higher than 0.7, which means that there is good reliability of the scale. Regarding discriminant validity, the values of MSV are less than AVE. As a result, this proves that the model has good construct validity. In other words, the first-order CFA indicates that the scale has acceptable validity in estimating the intercultural competence of foreign students at the UD.

**Table 3.** Model validity measures—first-order CFA.

|  | CR | AVE | MSV | MaxR(H) | Attitude | Knowledge | Awareness | Skills |
|---|---|---|---|---|---|---|---|---|
| **Attitude** | 0.960 | 0.670 | 0.575 | 0.966 | **0.818** | | | |
| **Knowledge** | 0.934 | 0.563 | 0.498 | 0.936 | 0.705 * | **0.750** | | |
| **Awareness** | 0.968 | 0.631 | 0.526 | 0.971 | 0.666 * | 0.584 * | **0.794** | |
| **Skills** | 0.936 | 0.649 | 0.575 | 0.940 | 0.759 * | 0.631 * | 0.725 * | **0.806** |

Note: there is no validity concerns. Significance of correlations: * $p < 0.001$. [86].

Besides that, it is possible to investigate the discriminant validity of this model using another approach called the heterotrait-monotrait (HTMT) ratio of correlations method (see the results in Table 4). According to [86], the threshold is 0.85 for strict discriminant validity. Accordingly, since the values are less than 0.85, it is possible to conclude that discriminant validity really exists between the constructs and their items.

**Table 4.** HTMT analysis—first-order CFA.

|  | Attitude | Knowledge | Awareness | Skills |
|---|---|---|---|---|
| Attitude | | | | |
| Knowledge | 0.714 | | | |
| Awareness | 0.674 | 0.585 | | |
| Skills | 0.763 | 0.637 | 0.726 | |

Note: there are no warnings for this HTMT analysis.

### 4.3.2. Second-Order CFA

According to the literature, intercultural competence dimensions are knowledge, attitude, skills, and awareness. Therefore, intercultural competence can be considered a general factor. In the context of CFA, it is possible to introduce intercultural competence to first-order CFA as a second-order factor (also called a higher-order factor). This can be achieved by removing the covariances between first-order factors and add paths, as well as errors for each of the latent variables, between general factors and first-order factors as shown in Figure 5.

As observed in Figure 5, in comparison with the first-order CFA, there were no required modifications. For instance, the goodness of fit indices of the second-order CFA matches the acceptable level. Furthermore, the goodness of fit indices of the two models are close enough to each other, as shown in Table 5. In addition, the factor loading of IC on knowledge, skills, attitude, and awareness are 0.76 ***, 0.87 ***, 0.87 ***, and 0.79, respectively (where *** $p < 0.001$). Additionally, the R-square for knowledge, skills, attitude, and awareness is acceptable (0.58, 0.76, 0.77, and 0.62). Hereby, the theory that intercultural competence includes four sub-constructs is well acceptable.

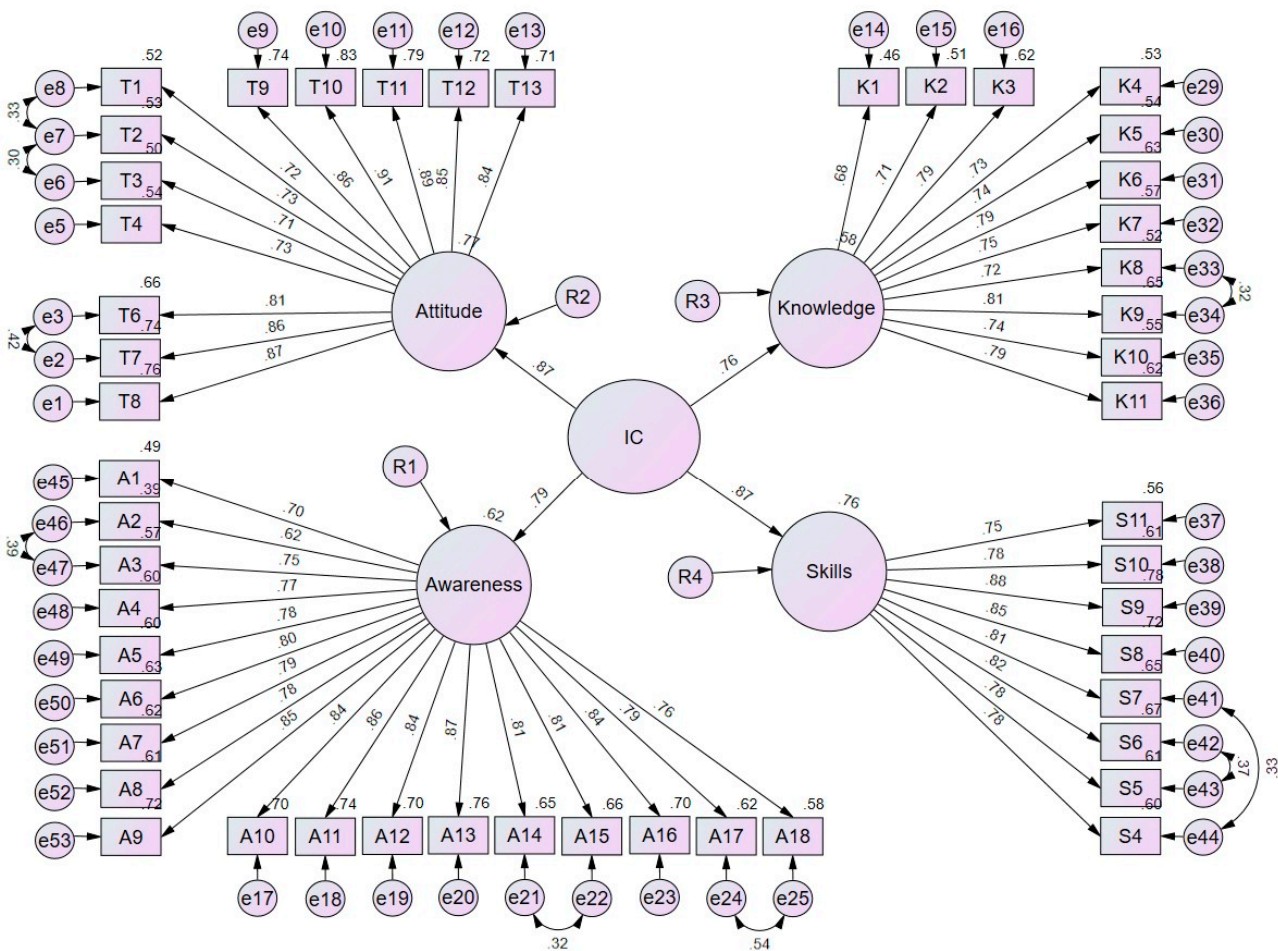

**Figure 5.** Second-order CFA (*n* = 384) with standardized estimates. Note: intercultural competence is the higher-order factor (indicated by IC).

**Table 5.** Goodness of fit indices—second-order CFA.

| Fit Indices | Value | Interpretation |
|---|---|---|
| Chi-square/*p*-value | 2580.660/*p* = 0.000 | |
| DF | 1114.000 | |
| CMIN/DF | 2.317 | Excellent |
| RMSEA | 0.059 | Excellent |
| CFI | 0.915 | Acceptable |
| SRMR | 0.044 | Excellent |

Note: as previously mentioned, the cut-off values for goodness of fit are as follows, RMSEA < 0.06, CFI > 0.95, CMIN/DF, between 1 and 3, and SRMR ≤ 0.08 [86].

As for validity and reliability for second-order CFA, AVE is higher than 0.5. As a result, the model has good convergent validity. Additionally, CR is higher than 0.7, which indicates good reliability of the scale. As for discriminant validity, the value of MSV is less than AVE. Based on that, the construct validity of this model is good in this case (see Table 6).

**Table 6.** Model validity measures—second-order CFA.

| | CR | AVE | MSV | MaxR(H) | IC |
|---|---|---|---|---|---|
| IC | 0.895 | 0.682 | 0.000 | 0.905 | **0.826** |

Note: there is no validity concerns here. Significance of correlations: *p* < 0.05 [86].

### 4.4. Structural Equation Model—KAS+A Variables

Because the goodness of fit indices as well as validity and reliability tests of first-order CFA have achieved an acceptable level, it is possible to develop SEM, which can be created by the rearrangement of first-order CFA. This rearrangement was conducted according to the previously mentioned theory in the literature section.

According to Fantini's theory, the development of knowledge, attitude, and skills has a positive impact on awareness. Furthermore, enhanced awareness helps to improve these factors. Hereby, in this case, awareness is considered an endogenous variable while knowledge, attitude, and skills are exogenous variables. These relationships are depicted in Figure 6.

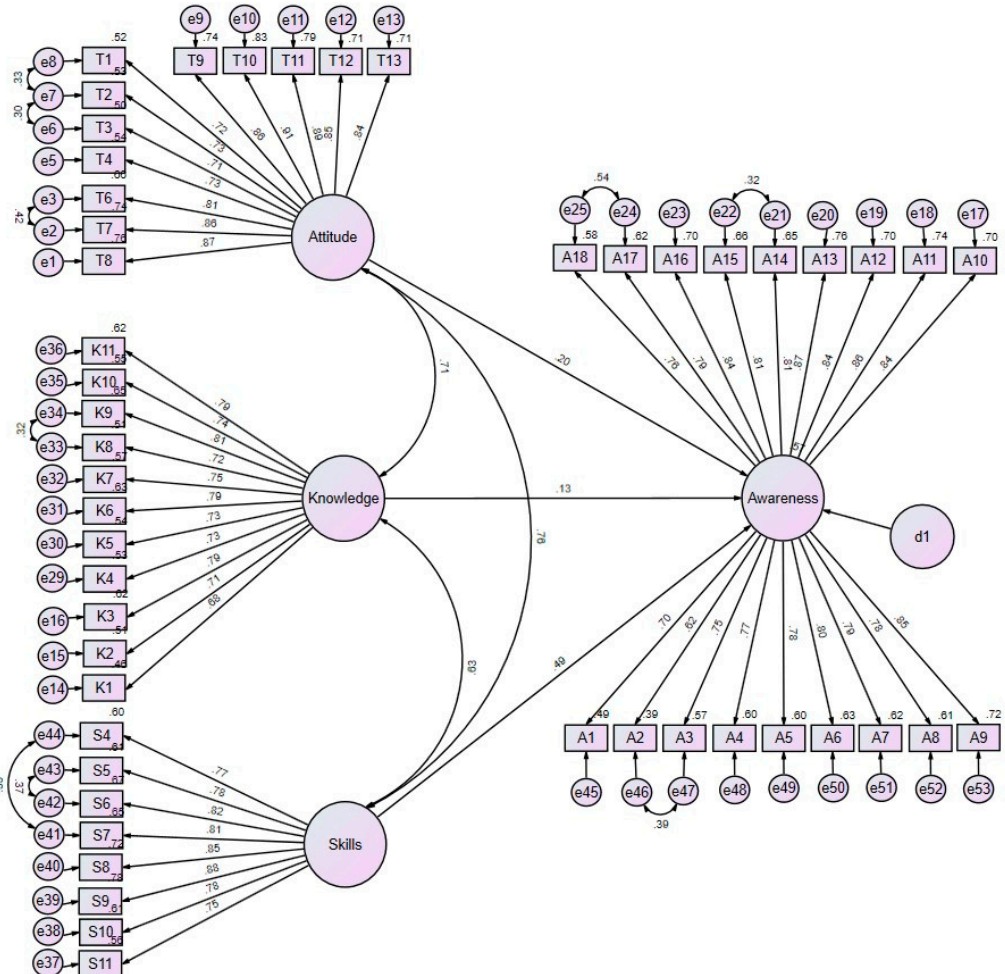

**Figure 6.** Second-order CFA (*n* = 384) with standardized estimates. Note: intercultural competence is the higher-order factor (indicated by IC).

Regarding direct effects in this model, the outcome is presented in Table 7. Based on the results, it is found that the skills factor has a highly positive effect on the awareness at the significance level of 0.001 (β = 0.487). Attitude also has a positive impact on awareness at a significance level of 0.010 (β = 0.204). Finally, there is a positive relationship between knowledge and awareness at a significance level of 0.050 (β = 0.133). As a result, the theory has been confirmed in this case study.

**Table 7.** Standardized direct effects—SEM.

| Predictor | Outcome | Std Beta |
|:---:|:---:|:---:|
| Attitude | Awareness | 0.204 ** |
| Knowledge | Awareness | 0.133 * |
| Skills | Awareness | 0.487 *** |

Note: where * $p < 0.050$, ** $p < 0.010$, and *** $p < 0.001$.

As for indirect effects, it can be found between awareness's observed variables and attitude, knowledge, and skills. Hereby, the indirect effects can be calculated by averaging the effects of awareness's observed variables on knowledge, attitude, and awareness. As a result, awareness has a positive indirect impact on knowledge, skills, and attitude (average $\beta = 0.105$, 0.385, and 0.151 respectively). These results also indicate that enhanced awareness plays an important role in the development of knowledge, skills, and attitude.

### 4.5. Foreign Students' Feedback—Open-Ended Question

To make the study more valuable, it was necessary to provide space by which respondents can share their thoughts, ideas, or feedback regarding the intercultural activities and programs conducted by the university. In this regard, the survey was ended with an open-ended question. After the exclusion of the missing data in this question, the responses are summarized as follows. Firstly, most respondents attribute the lack of intercultural communication to the language barrier problem, where it is not easy to learn the Hungarian language. Other respondents added cultural barriers as a part of the problem. Secondly, some respondents argue that the host culture lacks an interactive approach toward foreigners. Thirdly, others claim that awareness can be enhanced by intercultural experiences which are necessary for shaping positive interaction. Furthermore, one of the respondents emphasizes that the empathy factor plays an important role in intercultural communication.

### 5. Discussion

The main aim of this paper was to validate Fantini's intercultural competence scale for the foreign students at the UD. Starting with the exploratory factor analysis, four factors were extracted (intercultural knowledge, attitude, skills, and awareness). Besides that, the results proved that the scale has good reliability in estimating the intercultural competence for these data. Using a confirmatory factor analysis approach, first-order CFA indicates that the data fit the hypothesized model since the analysis confirmed that the model of this study has good construct validity (convergent and discriminant validity) as well as acceptable goodness of fit indices. As result, these findings confirm that Fantini's model is a multidimensional instrument consisting of the main four dimensions (KAS+A, knowledge, attitude, skills, and awareness). Based on that, it was possible to introduce a second-order factor to the model in which intercultural competence (as a general factor) loads on the four dimensions, with loading values ranging from 0.76 to 0.87. In addition, its construct validity, reliability, and goodness of fit indices have reached an acceptable level. Therefore, the second-order CFA confirms that intercultural competence is the general factor that accounts for the variation among the first-order factors (IC's main four dimensions) [29,61,82].

Finally, using a first-order CFA as a baseline model, it was possible to develop a structural equation model to investigate whether there is a casual relationship between the endogenous variable (awareness) and the exogenous variables (knowledge, attitude, and skills) or not. Based on the results, it was found that knowledge, attitude, and skills have a positive impact on the awareness of foreign students. In addition, enhanced awareness helps to promote the development of these variables. Therefore, intercultural awareness is considered a central component in this model and an integral part of the success of intercultural communication between different cultures [63,64,92,93]. Additionally, by looking at the results of SEM, students' intercultural skills have the highest significant impact on awareness. This is followed by attitude and knowledge in terms of the size of the influence.

Consequently, in comparison with other factors, foreign students' intercultural knowledge has the lowest impact on their intercultural awareness. According to the literature, having knowledge of the culture of the host country as well as their language helps to promote people's cross-cultural adaptation. Based on that, enhancing foreign students' intercultural knowledge can be achieved through the adoption of an integrative approach held by the university [94]. Therefore, the UD's responsibility is to afford the chance and opportunity which encourage and motivate foreign students to integrate into the host society. This procedure is considered a valuable source of culture as well as language learning [95]. By looking at the Hungarian literature, according to the study in [27], it was argued that there is a lack of opportunities for integration into the Hungarian higher education system. Therefore, foreign and local students do not participate and communicate with each other. Furthermore, it was found that both foreign and local students are interested in having common programs as well as joint courses. In addition, the events lack effectiveness in mixing the two groups [27]. This also matches the responses of participants to the open-ended question where several respondents claim that the interactive approach is still missing in the university environment.

On the other hand, most of the respondents attribute the lack of intercultural communication to the language barrier. About 87.24% of the respondents confirmed that they have experienced linguistic challenges to communicate with Hungarians. This was also observed in the answers of the participants to the open-ended question. Another challenge is related to the willingness of foreign students to learn the Hungarian language. About 50.90% of the respondents confirmed that they have joined Hungarian language courses. However, this percentage is not enough. Besides that, the lack of intercultural communication can be justified by the introverted personality of Hungarian students as well as language anxiety due to their weak English language competency [80]. This also matches the findings in [27,81]. According to Fantini, second language proficiency plays an important role in the success of intercultural communication. Therefore, having knowledge of the language of the other cultures is also critical for the success of the integration of the two groups. Regarding the Hungarian education system, it was found that there is a conflict between the policy and practice in teaching the English language in the Hungarian schools. In other words, in practice, language teachers mostly use their mother tongue in teaching the second language in the classroom. However, according to the policy, the teacher should concentrate on using the target language to enhance the communication as well as listening skills of Hungarian students [96]. This partly justifies why Hungarian students are hesitant to contact other cultures. On the other side, based on the feedback of the respondents, some argue that the Hungarian language is very difficult to be learned. This may discourage foreign students from learning the language and interacting with the host society.

As for intercultural attitude, the results indicate that foreign students' attitude positively affects their intercultural awareness (β = 0.204 **, at a significance level of 0.01). According to the authors of [97], there is an association between intercultural communication and the decline of anxiety as well as the rise of people's positive intercultural attitudes. Therefore, the UD should re-evaluate its current programs and undertake a new integrative approach for its students. This may include the development of policies and institutions which help to promote intercultural contact between foreign and local students. Intercultural contact and experience are necessary for the development of intercultural competence including intercultural knowledge, attitude, skills, awareness, and language proficiency for the two groups.

Finally, it was important to clarify that the study has successfully validated a western-developed model for a post-communist country (Hungary). Regarding policy implications, the literature includes many cases partially similar to the approach of this study. For instance, the work of [98] aimed to investigate the relationship between mediated contact and the global competence of Chinese university students. The study has used three dimensions developed in [99] (global attitudes, skills, and knowledge). Based on this

study, the university's decision-makers should provide students with face-to-face contact chances. This interactive approach could reduce the level of anxiety and enhance Chinese students' global competence. Hereby, this approach helps to encourage students to easily contact other cultures. In addition, the work in [100] has used the model of [101] as part of their study in order to investigate the relationship between intercultural competence (knowledge, attitude, skills, and awareness), intercultural experience, and the creativity of the Russian students in Moscow. The results of this study indicate that students' positive attitude toward other cultures, as well as intercultural experiences, has a positive influence on creativity. Finally, the study in [102] has used the model from [103] in order to investigate the relationship between sensation seeking, intercultural attitude, and intercultural communication competence. The study has targeted foreign students at the University Utara Malaysia. According to the results, it was recommended to afford foreign students the necessary training sessions as well as workshops in order to promote their intercultural competence.

The limitations of this study can be summarized as follows. Firstly, it was not possible to use awareness as a mediator since this model does not treat intercultural competence as a separate construct. Instead, the study just introduces intercultural competence as a general factor (using a higher-order CFA). Secondly, the circulation of the survey was not easy, and the progress was very slow due to the COVID-19 pandemic in that period as well as the length of the survey.

Regarding future studies, it is recommended to conduct a comparative study to figure out the role of each university in enhancing intercultural contact among foreign and Hungarian students. In addition, since the study emphasizes the importance of the integrative approach, it is recommended to investigate the impact of Fantini's KAS+A variables on the integration of foreign students into the host culture. This can be achieved by finding the suitable scale measuring integration factor and introducing it into the model as an endogenous variable.

## 6. Conclusions

This study addresses the internationalization performance of Hungarian higher education from a specific point of view. In fact, the variety of scholarship programs in Hungary played an important role in increasing the number of foreign students and enhancing the internationalization process of its entities. However, there are a number of obstacles that could hinder this process. For example, one of these obstacles could be the lack of an integrative approach in Hungarian universities, which could result in negative consequences on the intercultural interaction between foreign and Hungarian students. These types of obstacles could hinder the development of the foreign student's cognitive skills such as their intercultural knowledge about the host culture, skills on how to understand other cultures, attitude toward diversity, and the development of their awareness. On the other hand, the cognitive skills of Hungarian students could be negatively influenced as well. Other obstacles can be attributed to Hungarian students' introverted personalities as well as linguistic anxiety. Therefore, Hungarian universities should find a specific approach to solve these issues since intercultural competence becomes a necessary requirement for multinational firms in the labor market. Furthermore, achieving this could enhance the creativity of students at the university level [100] and decrease anxiety among them [98].

This study chose the University of Debrecen as part of the investigation for two reasons. Firstly, the University of Debrecen is well known as the biggest educational entity on the level of the country. Secondly, compared to other universities, it has the highest quota of international students. Therefore, it could be challenging to address this type of issue and satisfy the market needs.

By looking at the findings of this study, it is important to illustrate the fact that the model of Fantini is validated in a new environment (at a central European university). The validity and reliability of the model confirm the theory developed by Fantini, in which the intercultural competence of foreign students is represented by their intercultural knowledge,

skills, attitude, and awareness. Furthermore, based on the theory, the development of the intercultural awareness of foreign students can be boosted by enhancing intercultural skill, knowledge, and attitude. In other words, if the foreign student has good skills to analyze and learn about other cultures, have high openness and empathy toward other cultures, and sufficient knowledge about his/her own culture and others, this means that he/she has a good level of intercultural awareness by which he/she can understand and compare his/her own culture with others. Similarly, having a high level of intercultural awareness could help to improve the rest of these three factors. However, compared to other dimensions, the intercultural knowledge of foreign students has the least effect on their intercultural awareness. The justification behind this problem is attributed to the absence of opportunity by which intercultural interaction could take place. Additionally, the university does not provide preparatory training courses at the beginning of each semester which could prepare the students to know about the host culture and settle easily. Based on that, the intercultural knowledge of foreign students about the host culture is not sufficient. On the other hand, foreign students provided useful feedback, which matches evidence provided by Hungarian literature, on the necessity of applying integrative methods by the host university such as arranging joint courses for Hungarian and foreign students, common events, and the enhancement of linguistic skills of the two parties. In addition, most of UD's foreign students argued that linguistic and cultural barriers represent significant obstacles hindering intercultural interaction. Others also emphasize the importance of an interactive approach by the host university to ease intercultural communication, which is useful for the development of their intercultural knowledge about the host culture [104].

Based on the previous summary, the policy implication of this study can be listed as follows. Firstly, to satisfy the needs of the labor market, the intervention of the UD should take place to improve the intercultural competence of the international as well as Hungarian students. Secondly, the UD should regularly arrange common events which aim to consider the participation of the international and Hungarian students. This type of event could enhance the intercultural interaction of the students and improve their intercultural awareness. Thirdly, at the beginning of each semester, the UD should afford preparatory courses for the newcomers and current students. These courses aim to enhance their intercultural knowledge of the host culture and prepare them to perform effectively in different communication scenarios. This could remove the anxiety and stress among international students while they are socializing with Hungarian students. Fourthly, since both parties highlight the existence of a linguistic barrier, the university should force a policy that forces international students to learn the language of the host culture. Learning the language of the host culture could ease the process of integration and avoid any misunderstanding.

**Author Contributions:** J.T. and O.A. designed the model and the computational framework and analyzed the data. O.A. and P.B. carried out the implementation. J.T. performed the calculations. J.T. and O.A. wrote the manuscript with input from all authors. P.B. and P.L. conceived the study and were in charge of overall direction and planning. All authors have read and agreed to the published version of the manuscript.

**Funding:** This research received no external funding.

**Institutional Review Board Statement:** Not applicable.

**Informed Consent Statement:** Not applicable.

**Data Availability Statement:** Not applicable.

**Conflicts of Interest:** The authors declare no conflict of interest.

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
