# Peer review of "Psychometric Properties of Intercultural Competences in a Central European Context"

_sustainability, doi:10.3390/su14127502_

Round 1

Reviewer 1 Report

Research is relevant and interesting.

Author Response

We highly appreciate your positive opinion.

Thank you very much!

Reviewer 2 Report

This is an important study of an attempt to validate a a prior assessment tool of intercultural competence. The study seems to be original and planned well. The research and reporting, for the most part, is sound. However, there are some minor gaps in the cited literature. Thus, the authors could use some of the following studies that they would deem appropriate and relevant to diversify the references:

·        Çelik, S. (2021). Culture and language instruction: Does Turkey's EFL curriculum do enough to support intercultural awareness? International Journal of Curriculum and Instruction, 13(3), 3443–3463.

·        Müller, F., Denk, A., Lubaway, E., Sälzer, C., Kozina, A., Perše, T. V., Rasmusson, M., Jugović, I., Lund Nielsen, B., Rozman, M., Ojsteršek, A., & Jurko, S. (2020). Assessing social, emotional, and intercultural competences of students and school staff: A systematic literature review. Educational Research Review, 29, 100304.

·        Çelik, S., & Erbay-Çetinkaya, Ş. (2020). Culture in English language teacher education programs. In Y. Bektaş Çetinkaya (Ed.), Intercultural competence in ELT (pp. 39-64). Peter Lang.

·        Smith, E. L. (2016). An examination of the psychometric properties of and validity evidence for the Alliant Intercultural Competency scale. Masters Theses, 2010-2019. 97.
https://commons.lib.jmu.edu/master201019/97

·        Adam, F., Makarovič, M., Rončević, B., & Tomšič, M. (2004). The challenges of sustained development: The role of socio–cultural factors in East Central Europe. Central European University Press.

·        Çelik, S. (2014). Classroom strategies of Turkish EFL teachers in managing cultural In P. Romanowski (Ed.), Intercultural issues in the era of globalization (pp. 32- 46). Wydawnictwo Naukowe.

The conclusion section is weak; the authors should implement strategies to help move beyond merely summarizing the introduction and/or key points of the research. Therefore, the authors should provide a brief summary of what was learned from the research, and then mainly focus on the implications, evaluations, insights, and final messages that they would like to offer.

Finally, although the mabuscript  is well written, the final text might use a careful proofreading and polishing to increase the readability. Overall, the manuscript has the potential to make a meaningful contribution to the field, as it substantiates the use of an instrument for assessing foreign students’ intercultural competence for policy implementations.

Author Response

  1. We highly appreciate your positive opinion in general. Thanks for your suggested studies, we used them.
  2. We have developed the conclusion chapter according to your suggestions.

Reviewer 3 Report

Overall great analysis but I am not sure about the first research question, which is to see whether the data fits the measurement model. 

Author Response

We highly appreciate your positive opinion in general. We agree with your opinion on the first question, so we have deleted it.

Reviewer 4 Report

1. In order to emphasize the originality of the paper, I would suggest the authors to improve the contribution section of their article, whether it will be given in the introduction or discussion.

2. What are  the implications of this study, try to celebrate your point of view in this regard

3. The theoretical framework of the study is at large is missing try to link a theory to your study, without a theory how it is possible to even make hypotheses.

4. Enhance knowledge on the conclusion it needs to be more relatable to the result of the research and it should be different from the introduction. 

Based on the above comments the paper needs improvements in order to make it more demanding. 

Author Response

  1. Thank you for your suggestion. We inserted the contribution section at the end of the introduction.
  2. Thank you for your comment. We wrote about the implications of this study.
  3. Thank you for your suggestion. We inserted a Theoretical framework section into the study, where we tried to link the theory to our study.
  4. We developed the conclusion section according to your comment.